# Targeting Mitochondria in Melanoma

**DOI:** 10.3390/biom10101395

**Published:** 2020-09-30

**Authors:** Sepideh Aminzadeh-Gohari, Daniela D. Weber, Luca Catalano, René G. Feichtinger, Barbara Kofler, Roland Lang

**Affiliations:** 1Research Program for Receptor Biochemistry and Tumor Metabolism, Department of Pediatrics, University Hospital of the Paracelsus Medical University, 5020 Salzburg, Austria; s.aminzadeh-gohari@salk.at (S.A.-G.); d.weber@salk.at (D.D.W.); l.catalano@salk.at (L.C.); r.feichtinger@salk.at (R.G.F.); 2Department of Dermatology and Allergology, University Hospital of the Paracelsus Medical University, 5020 Salzburg, Austria

**Keywords:** melanoma, mitochondrial respiration, antibiotic, anti-parasitic drug, ONC212, Warburg effect, BRAF, NRAS

## Abstract

Drastically elevated glycolytic activity is a prominent metabolic feature of cancer cells. Until recently it was thought that tumor cells shift their entire energy production from oxidative phosphorylation (OXPHOS) to glycolysis. However, new evidence indicates that many cancer cells still have functional OXPHOS, despite their increased reliance on glycolysis. Growing pre-clinical and clinical evidence suggests that targeting mitochondrial metabolism has anti-cancer effects. Here, we analyzed mitochondrial respiration and the amount and activity of OXPHOS complexes in four melanoma cell lines and normal human dermal fibroblasts (HDFs) by Seahorse real-time cell metabolic analysis, immunoblotting, and spectrophotometry. We also tested three clinically approved antibiotics, one anti-parasitic drug (pyrvinium pamoate), and a novel anti-cancer agent (ONC212) for effects on mitochondrial respiration and proliferation of melanoma cells and HDFs. We found that three of the four melanoma cell lines have elevated glycolysis as well as OXPHOS, but contain dysfunctional mitochondria. The antibiotics produced different effects on the melanoma cells and HDFs. The anti-parasitic drug strongly inhibited respiration and proliferation of both the melanoma cells and HDFs. ONC212 reduced respiration in melanoma cells and HDFs, and inhibited the proliferation of melanoma cells. Our findings highlight ONC212 as a promising drug for targeting mitochondrial respiration in cancer.

## 1. Introduction

Accounting for ~4% of cancers in adolescents, cutaneous melanoma has become a common malignancy owing to its dramatic rise among fair-skinned people [1]. For decades, the prognosis of patients with melanoma was very poor once metastatic disease had developed. Approval in 2011 of the BRAF-inhibitor vemurafenib and the cytotoxic T-lymphocyte-associated protein 4 (CTLA-4)-antibody ipilimumab has resulted in significant improvements in the treatment of advanced melanoma. Similarly, the use of ‘checkpoint’ inhibitors to target programmed cell death protein 1 (PD-1), which is expressed on immune cells and triggers immunosuppressive signaling pathways [2], can overcome T-cell anergy and activate the anti-tumor immune response. Thus the introduction of anti–PD-1–based therapies, either in combination with ipilimumab or as single agents, has further improved the overall survival of patients with metastatic melanoma (reviewed in [3]). Despite the effectiveness of these treatments, approximately half of advanced melanoma patients still die from their disease, underlining the urgent need for additional therapeutic strategies. 

There has been a recent resurgence of interest in studying metabolic features of cancer cells and how alterations therein can be therapeutically targeted. In general, altered energy metabolism is considered a hallmark of cancer, which is linked to cancer metastasis, drug resistance, and survival rate [4]. Cancer cells, in contrast to non-cancerous cells, show fundamental metabolic changes, exhibiting higher glucose uptake and dependency on glycolysis, known as the Warburg effect. In early studies, impaired aerobic respiration was assumed to be the reason for the Warburg effect. However, increasing experimental evidence shows that many subsets of tumor types, such as melanoma, still maintain high levels of mitochondrial energy metabolism despite being glycolytic [5,6,7,8,9,10]. Upregulation of OXPHOS in melanoma contributes to tumor invasion, metastasis, and increased resistance to mitogen-activated protein kinase (MAPK) pathway inhibitors, one of the most common anti-melanoma therapies [7,10,11,12,13,14,15,16]. Activating mutations in BRAF and NRAS in the MAPK pathway have been identified in melanomas at frequencies of 50% and 20%, respectively [17,18]. This pathway contributes to melanoma development and plays a key role in metabolic changes in cancer cells [19,20]. Inhibition of BRAF leads to impairment of glycolysis and increased OXPHOS capacity [21,22,23,24,25]. BRAF-mutated melanoma cells depleted of mitochondrial DNA show higher sensitivity to BRAF inhibitors, highlighting a key role of mitochondria in the development of resistance to MAPK pathway inhibitors [26]. Melanin and intermediates involved in melanogenesis are also possible regulators of melanoma metabolism since they can switch the metabolism of normal and malignant melanocytes from aerobic to anaerobic glycolysis [27,28,29]. Melanogenesis also upregulates the expression of hypoxia-inducible factor-1α (HIF-1α) and HIF-1-dependent target genes involved in glucose metabolism [30]. However, in a recent study, we did not observe differences in HIF-1α protein levels among melanoma tissues [8]. In contrast, Slominski et al. reported higher levels of HIF-1α in advanced melanoma compared to thin melanoma localized in the skin [29,30].

Taken together, mitochondrial respiration seems to be a causative factor in tumor progression and therapy resistance in melanoma [9,12,15,16]. Therefore, targeting mitochondrial respiration might offer novel therapeutic strategies against melanoma [13,16]. Various drugs are reported to influence mitochondrial respiration [31]. A growing number of pre-clinical and clinical studies indicates that antibiotics like tigecycline (TIG), doxycycline (DOX) and azithromycin (AZI) as well as the anti-parasitic drug pyrvinium pamoate (PP) have the potential to inhibit the growth of a wide spectrum of solid tumors and blood cancers. These drugs impact mitochondrial functions, for example by reducing mitochondrial biogenesis, altering mitochondrial morphology, lowering OXPHOS activity and increasing the level of reactive oxygen species [31,32,33,34,35,36,37,38]. In vitro analysis showed that cancer cells depleted of mitochondrial DNA develop resistance to these types of antimicrobial drugs, indicating that mitochondria are specific targets of such treatments [31]. Newly developed anti-tumor agents like the imipridone family including ONC201, ONC206, and ONC212 demonstrate anti-tumor effects in a variety of cancers. Remarkably, they had no effects on non-cancerous cells [39,40,41]. Recently, suppression of mitochondrial respiration has been reported as a novel mechanism of action of imipridone agents in glioblastoma and breast cancer [42,43,44]. 

In this article, we first investigated the level of OXPHOS and glycolysis in melanoma cells bearing different genetic alterations. We then explored the effects of TIG, DOX, AZI, PP and ONC212 on melanoma proliferation and mitochondrial respiration in vitro. 

## 2. Materials and Methods 

### 2.1. Cell Lines, Primary Cells and Agents

The melanoma cell lines WM3311 (BRAF/NRAS wild-type; Wistar Institute, Philadelphia, PA, USA), WM3000 (BRAFwt/NRASQ61R; Rockland Inc., Philadelphia, PA, USA), WM47 (BRAFV600E/NRASwt; Wistar Institute, Philadelphia, PA, USA) were cultured in MCDB153 Medium (Sigma Aldrich, Darmstadt, Germany) supplemented with 20% Leibovitz’s L-15 Medium (Sigma Aldrich, Darmstadt, Germany), 10% fetal bovine serum (FBS) (Gibco, Darmstadt, Germany), 1% Penicillin-Streptomycin-Amphotericin mixture (Lonza, Basel, Switzerland), 1.68 mM CaCl_2_ (Sigma Aldrich, Darmstadt, Germany) and 5 µg/mL insulin (Sigma Aldrich, Darmstadt, Germany). The melanoma cell lines WM3311 and WM47 were kindly provided by Dr. Meenhard Herlyn, Wistar Institute. A375 (BRAFV600E/NRASwt; Sigma Aldrich, Darmstadt, Germany) melanoma cells and primary human dermal fibroblasts (HDFs) (kindly provided by Prof. Johannes Mayr, Department of Pediatrics, University Hospital of the Paracelsus Medical University Salzburg) were cultured in DMEM-high glucose (Sigma Aldrich, Darmstadt, Germany) supplemented with 10% FBS and 1% Penicillin-Streptomycin-Amphotericin mixture. All cells were kept at 37 °C in a 5% CO_2_ atmosphere. We generated vemurafenib-resistant A375 cells (RA375) through continuous treatment of vemurafenib-sensitive A375 cells with 3 µM vemurafenib for 5 weeks (Appendix A). Vemurafenib-resistant A375 cells were maintained in medium containing 3 µM vemurafenib. Vemurafenib (PLX4032), TIG, DOX, AZI, and ONC212 were purchased from Selleckchem (Houston, TX, USA).

### 2.2. Proliferation Assays

Cell viability was measured by MTT (3-(4,5 Dimethylthiazol-2-yl)-2,5-Diphenyltetrazolium Bromide) assay (Sigma Aldrich, Darmstadt, Germany). Cells were seeded into 96-well plates (2000–4000 cells per well). After 24 h, the cultures were supplemented with 100 µM, 30 µM, 10 µM, and 3 µM TIG, DOX, or AZI, as well as 1000 nM, 300 nM, 100 nM, and 30 nM PP or ONC212. The highest concentration of DMSO applied in the drug tests was used as the vehicle control. The total length of the treatments was 3 days. The percentage of surviving cells was calculated and the half maximal inhibitory concentration (IC50) was determined by log (inhibitor) compared to response—variable slope, using GraphPad Prism 8.

### 2.3. Spheroids

Spheroids were generated from two melanoma cell lines (WM47 and WM3000) and two HDFs by seeding 4000–6000 cells in 1% agarose-coated 96-well flat bottom plates. Three days after seeding, spheroids were treated with the drugs for five days. On day 3, the medium including drugs was exchanged. Images were taken at days 1 and 5 after treatment of the spheroids.

### 2.4. Western Blot

Snap-frozen cell pellets (~50 mg) were homogenized in 5-fold volume extraction buffer (20 mM Tris-HCl, pH 7.6, 250 mM sucrose, 40 mM KCl, 2 mM EGTA). The post-nuclear supernatant (600 g homogenate) containing the mitochondrial fraction was used for western blot analysis with the following antibodies: Anti-VDAC1 (1:2000, ab15895; Abcam, Cambridge, UK), Anti-NDUFS4 (1:1500, ab55540; Abcam), Anti-SDHA (1:2000, ab14715; Abcam), Anti-UQCRC2 (1:1500, ab14745; Abcam), Anti-MTCO2 (1:2000, ab79393; Abcam), Anti- ATP5A (1:2000, ab110273; Abcam), and Anti-Vinculin (1:10,000, ab129002; Abcam). The visualized bands were quantified with Bio-Rad’s Image Lab software. The intensity of the bands was normalized to that of vinculin. 

### 2.5. Enzymatic Activities

Citrate synthase (CS) and OXPHOS enzyme activities were determined by spectrophotometric measurements as previously described [45]. The 600 g homogenate was used to measure OXPHOS enzyme activities in four melanoma cell lines and one HDF line. Furthermore, isolated mitochondria from RA375, A375, ONC212-treated A375 and vehicle treated-A375 pellets were used for enzymatic measurements. Mitochondria were isolated according to Bentlage et al. [46]. In brief, citrate synthase was measured according to Srere et al. [47] with modifications. The reaction mixture contained 50 mM Tris-HCl pH 8.1, 0.1% bovine serum albumin (BSA), 0.1% Triton X-100, 0.2 mM 5,5’-dithio-bis (2-nitrobenzoic acid) and 0.15 mM acetyl-CoA, and 10 µL of 600 g homogenate or 2 µL of isolated mitochondria. After initially recording thiolase activity for 3 min, the citrate synthase reaction was started by addition of 0.5 mM oxaloacetate and was followed at 412 nm for 8 min.

The activity of NADH/decylubiquinone oxidoreductase (complex I) was measured by monitoring NADH oxidation in the presence of quinone at 340 nm for 4 min. After 3 min pre-incubation of samples (10 µL of 600 g homogenate or 10 µL isolated mitochondria) at 37 degrees Celsius, the reaction was started by the addition of 0.2 mM NADH. Complex I activity was calculated using the difference between the measured rates in the absence vs. presence of 0.5 mg/mL rotenone, an inhibitor of complex I.

The activity of succinate:ubiquinone-oxidoreductase (complex II) was measured according to Rustin et al. [48] with the following modifications. The reaction mixture contained 50 mM potassium phosphate pH 7.8, 2 mM ethylenediaminetetraacetic acid (EDTA), 0.1% BSA, 3 µM rotenone, 80 µM 2,6-dichlorophenol, 50 µM decylubiquinone, 1 µM antimycin A, 0.2 mM ATP, 0.3 mM KCN, and 10 µL of 600 g homogenate or 2 µL of isolated mitochondria. The mixture was preincubated for 10 min at 37 degrees Celsius, the reaction was started by addition of 10 mM succinate, and followed for 6 min at 600 nm.

The activity of coenzyme Q/cytochrome c—oxidoreductase (complex III) was determined in the presence of 50 mM potassium phosphate buffer pH 7.8, 2 mM EDTA pH 8.6, 0.3 mM KCN, 100 µM cytochrome c, 200 µM reduced decyl-ubiquinol. The reaction was started by the addition of samples (10 µL of 600 g homogenate or 2 µL of isolated mitochondria). The reduction of cytochrome c was monitored over 3–4 min at 550 nm. The reaction was inhibited by the addition of 1 μM antimycin. Antimycin A-insensitive activity was subtracted from the total activity to calculate complex III activity.

The enzyme activities of ferrocytochrome c/oxygen oxidoreductase (complex IV), and the oligomycin-sensitive ATPase activity of the F1F0 ATP synthase (complex V) were determined by using buffer conditions as previously described by Rustin et al. [48]. The complex IV activity was measured by following the oxidation of cytochrome c at 550 nm for 6 min. 10 µL of 600 g homogenate or 2 µL of isolated mitochondria were used. The whole reaction mixture for the ATPase activity measurement was treated for 10 s with an ultra-sonifier (Bio cell disruptor 250, Branson, Vienna, Austria). The complex V activity was measured by following the oxidation of NADH at 340 nm for 4 min. 7 µL of 600 g homogenate or 2 µL of isolated mitochondria were used. All spectrophotometric measurements (Uvicon 922, Kontron, Milan, Italy) were performed at 37 degrees Celsius. All reagents were obtained from Sigma Aldrich (Darmstadt, Germany). 

### 2.6. Bioenergetic Measurements

Cellular bioenergetic measurements were performed using the Seahorse XFe96 Analyzer (Agilent Technologies). For the metabolic characterization of the melanoma cell lines, cells were seeded in XF96 cell culture microplates (Seahorse Biosciences) at a density of 20,000 cells/well one day prior to the analysis. For drug treatment studies, cells were seeded at a density of 5000–10,000 cells/well and treated for 24 h with the respective drugs. On the day of the assay, cell culture medium was removed and replaced by assay medium (Seahorse XF DMEM medium) supplemented with 10 mM glucose, 1 mM sodium pyruvate and 2 mM L-glutamine for the XF Cell Mito Stress Test, whereas the assay medium for the XF Glycolysis Stress Test was supplemented with 1 mM sodium pyruvate and 2 mM L-glutamine only. To determine mitochondrial flux using the XF Cell Mito Stress Test, basal cellular oxygen consumption was determined, followed by sequential injection of the ATP synthase inhibitor oligomycin (5 μM; port A), the uncoupler fluoro-carbonyl cyanide phenylhydrazone (FCCP) (2 μM; port B), and the complex I inhibitor rotenone (0.5 μM; Port C) combined with the complex III inhibitor antimycin A (0.5 μM; port C). For the evaluation of glycolytic flux using the XF Glycolysis Stress Test, non-glycolytic acidification was determined, followed by sequential injection of glucose (10 mM; Port A) to evaluate basal glycolysis, the ATP synthase inhibitor oligomycin (5 μM; port B), and the hexokinase inhibitor 2-deoxy-glucose (50 mM; port C). Each measurement was normalized to cell content, quantified by crystal violet assay (Sigma-Aldrich) as performed previously [49].

### 2.7. Statistical Analyses

All statistical analyses were performed by using GraphPad Prism 8. Data are presented as the mean ± standard deviation from 2 or 3 independent experiments with two to four replicates.

## 3. Results

### 3.1. Melanoma Cells Exhibit Elevated Levels of Glycolysis and OXPHOS, but Contain Dysfunctional Mitochondria 

It is well accepted that many tumor cells depend almost exclusively on glycolytic metabolism for producing energy, whereas non-cancerous cells mostly rely on mitochondrial respiration. To gain insight into the levels of OXPHOS and glycolysis in melanoma cells, we initially evaluated the levels of OXPHOS complexes and voltage-dependent anion-selective channel 1 (VDAC1) (as a marker for mitochondrial mass) in four melanoma cell lines with mutations affecting *BRAF* or *NRAS* or neither oncogene—BRAF and NRAS wild-type (WM3311), BRAF-mutated (WM47 and A375), and NRAS-mutated (WM3000)—and compared them to HDFs as non-cancerous skin cells. Based on Western blot analysis, the expression of VDAC1 and OXPHOS complexes was generally higher in the melanoma cell lines WM3311, WM47, and A375 compared to WM3000 and HDFs (Figure 1a–g). 

We also measured the activity of citrate synthase (another marker for mitochondrial mass) and the OXPHOS complexes to examine the functional level of the OXPHOS complexes. The enzymatic measurements showed a similar pattern as the western blot analysis. The activity of citrate synthase and the OXPHOS complexes was higher in WM3311, WM47, and A375 compared to WM3000 and HDFs (Figure 1h–m). 

Next, we performed a series of real-time measurements of oxygen consumption rate (OCR), an indicator of OXPHOS activity, and extracellular acidification rate (ECAR), an indicator of glycolytic rate, using the Seahorse XF Cell Mito Stress and Glycolysis Stress tests (Figure 2). The melanoma cell lines WM3311, WM47, and A375 showed higher levels of basal glycolysis and basal respiration compared to WM3000 and HDFs (Figure 2a,c,b,h). The level of maximal respiration (MAXR), an indicator of reserved respiratory capacity, after injection of the uncoupler fluoro-carbonyl cyanide phenylhydrazone (FCCP) was higher in WM3311, WM47 and A375 compared to WM3000 and HDFs (Figure 2a,d). In agreement with the higher OCR, the level of ATP-linked respiration (ATP-R) was significantly higher in WM3311, WM47, and A375 compared to HDFs, and even lower in WM3000 (Figure 2e). The parameter equations from the Mito Stress test revealed remarkably higher proton leakage across the mitochondrial membrane and lower mitochondrial coupling efficiency in all melanoma cell lines compared to HDFs, clearly indicating a dysfunction in mitochondrial respiration (Figure 2f,g). The Glycolysis Stress test revealed significantly higher basal glycolysis and maximal glycolytic capacity (measured after mitochondrial ATP synthase inhibition) of melanoma cells compared to HDFs (Figure 2b,h,i). The glycolytic reserve, which indicates the discrepancy between maximal and basal glycolytic flux (or, in other words, how close the glycolytic function is to the cell’s theoretical maximum) was higher in WM3311, WM47, and A375 compared to WM3000 and HDFs (Figure 2j). Overall, the ratio of respiration to glycolysis was very low in all the melanoma cell lines compared to HDFs (Figure 2k), again indicating higher dependency on glycolysis of the melanoma cell lines.

### 3.2. OXPHOS Activity in BRAF Inhibitor-Resistant A375 Cells 

Several studies showed upregulation of OXPHOS in BRAF inhibitor-resistant melanoma cells, suggesting that targeting mitochondrial respiration could be an effective strategy to overcome MAPK inhibitor resistance [13,16,26,50,51]. In this context, we measured the activity of OXPHOS complexes in isolated mitochondria from vemurafenib-resistant RA375 cells compared to parental A375 cells. The activities of all OXPHOS complexes were comparable between RA375 and A375, except complex IV, whose activity was significantly higher in RA375 cells (Figure 3a–f).

### 3.3. Effects of Mitochondrial Inhibitors on Respiratory Parameters of Melanoma Cells 

Since the melanoma cell lines are characterized by high OXPHOS, and therefore high mitochondrial respiration, drugs inhibiting respiration might impair melanoma energy metabolism and suppress tumor cell proliferation. To begin to test this potential, we treated melanoma cell lines and HDF control cells for 24 h with 30 µM TIG, 30 µM DOX, 30 µM AZI, 300 nM PP, or 300 nM ONC212 and measured mitochondrial respiration using the Seahorse XF Cell Mito Stress test. 

The antibiotics TIG, DOX and AZI have been shown to inhibit mitochondrial translation, thereby reducing the activities of OXPHOS complexes I, III, IV, and V, which contain mitochondrially encoded subunits [31]. Treatment of A375 and WM3000 cells with TIG, DOX, or AZI led to a 50% to 75% reduction in OCR and ATP-R levels. In contrast, WM47 and WM3311 cells were not affected (Figure 4a,d,f). However, TIG- and DOX-treatment of all melanoma cell lines decreased MAXR levels by 52% to 75% (Figure 4a,e). The MAXR of HDFs was not influenced by antibiotics, although the OCR and ATP-R were reduced especially by TIG (Figure 4a,d,e,f). In general, DOX had the strongest effect on MAXR, followed by TIG, whereas AZI had almost no effect. 

PP is known to suppress the NADH-fumarate reductase system and to lower mitochondrial respiration [52]. Treatment with 300 nM PP for 24 h resulted in a dramatic reduction in mitochondrial activity in all melanoma cell lines as well as HDFs (Figure 4b). Treatment of melanoma cells and HDFs with PP led to an approximately 80% reduction of basal OCR compared to vehicle treatment of cells. Unexpectedly, treatment with PP resulted in an increase of OCR after injection of oligomycin (an inhibitor of the ATP-synthase), suggesting that PP most likely affected mitochondrial membrane potential and led to uncoupled respiration (Figure 4b). The unusual increase of OCR after oligomycin injection did not allow calculation of Mito Stress Test Parameter equations for the PP treatment (Figure 4b).

ONC212 is reported to inhibit mitochondrial respiration in glioblastoma and breast cancer [42,43]. ONC212-treatment of melanoma cells (A375, WM47, and WM3000) led to 58% to 78% reductions in basal OCR and 55% to 89% lower ATP-R. Interestingly, basal OCR and ATP-R showed no sensitivity to ONC212 in WM3311 cells (Figure 4c,d,f). MAXR was reduced by ONC212 in all melanoma cell lines, ranging from 65% to 89% (Figure 4c,e). ONC212-treatment of HDFs also resulted in 80% to 50% lower basal OCR, 73% to 45% lower ATP-R, and 53% to 57% lower MAXR (Figure 4c–f). 

### 3.4. Effects of Mitochondrial Inhibitors on Melanoma Cell Proliferation 

Having demonstrated that drugs in our panel inhibit mitochondrial respiration in melanoma cells, we next explored whether such respiratory impairment suppresses cell proliferation. The anti-proliferative effect of drugs was measured after three days of treatment of the melanoma cell lines and HDFs with TIG, DOX, AZI (3, 10, 30, and 100 µM), PP, and ONC212 (30, 100, 300, and 1000 nM).

Of the three antibiotics, TIG was the most effective in suppressing growth of the melanoma cell lines in a dose-dependent manner, with a half maximal inhibitory concentration (IC50) ranging from 13 to 30 µM. TIG suppressed the proliferation of HDFs to a lesser extent, with IC50 values ranging from 145 to 265 µM (Figure 5). DOX was the second most effective antibiotic. The IC50 for DOX in WM3311, WM3000, and WM47 cells was slightly higher than for TIG (IC50 of 24–53 µM). Surprisingly, the A375 cell line was less sensitive to DOX (IC50 ~ 600 µM). HDFs were almost resistant to DOX, so that the highest concentration of DOX (100 µM) reduced the HDF proliferation by less than 20% (Figure 5). AZI had almost no effect on either the melanoma cell lines or HDFs, except at the highest concentrations (30 and/or 100 µM). PP dose-dependently inhibited proliferation of the melanoma cells and HDFs, with IC50 values of 109–379 nM and 510–815 nM, respectively (Figure 5). ONC212 reduced HDF growth slightly (by less than 20%), but most efficiently suppressed proliferation in all melanoma cell lines in the nanomolar range, IC50 204–786 nM. In addition, we tested the effect of ONC212 on vemurafenib-resistant A375 cells (RA375) compared to parental A375 cells. Similar to A375 cells, ONC212 was able to suppress the growth of R375 cells, with IC50 233 nM (Figure 5).

Since TIG, DOX, and ONC212 had less impact on the growth of HDFs compared to melanomas, we further tested the effect of these three drugs on melanoma-spheroids. Treatment of melanoma spheroids with 30 µM TIG, 30 µM DOX, or 300 nM ONC212 for five days caused size reductions of the spheroids compared to spheroids treated with vehicle. However, drug-treatment of HDF spheroids did not affect their size (Appendix A). 

### 3.5. ONC212 Reduces the Activity of OXPHOS Complexes

In light of the consistent influence of ONC212 on both respiratory parameters (OCR, ATP-R and MAXR) and proliferation in all melanoma cell lines, but not on HDFs, we further verified the direct effect of ONC212 on OXPHOS complex activity in isolated mitochondria from A375 cells treated for 24 h with 300 nM ONC212. ONC212-treatment of A375 lowered the activity of citrate synthase by 20%, of complex I by 60%, complex II by 41%, complex III by 63%, and complex IV by 30% (Figure 6a–f). Surprisingly, ONC212 had no effect on the activity of complex V (Figure 6g). 

## 4. Discussion

Insufficient respiration and ATP production due to mitochondrial dysfunction is considered to be the major cause of the Warburg effect. This view has recently been challenged by growing evidence that many cancer cells with sufficient mitochondrial respiration still retain elevated glycolytic levels. For instance, various studies report higher levels of both OXPHOS and glycolysis in melanoma cells compared to non-cancerous cells such as fibroblasts or melanocytes [8,53]. Consistent with these studies, a subset of melanomas that overexpresses microphthalmia transcription factor (MITF) and peroxisome proliferator-activated receptor gamma coactivator 1-alpha (PGC1α), a master regulator of mitochondria, showed upregulation of oxidative metabolism [54]. In agreement, we observed marked elevation of glycolysis in addition to high OXPHOS, as shown by higher basal OCR, ATP-R, and activity levels of OXPHOS complexes, in three out of four of our melanoma cell lines (a BRAF/NRAF/NF1-wild-type and two BRAF-mutated melanoma cell lines) compared to HDFs. A limitation of our study is the lack of comparison of our melanoma cell lines to melanocytes. 

Although the melanoma cell lines showed high respiration, they showed high uncoupled respiration. In addition, the elevated rate of respiration after injection of FCCP dropped in the melanoma cell lines, whereas it was stable in HDFs. Taken together, the observed lower coupling efficiency, reduction in FCCP-induced MAXR rate over time, and high level of proton leakage, all indicate mitochondrial dysfunction in our melanoma cells. These findings could reflect damaged electron transport proteins or OXPHOS complexes in melanoma cells [55,56]. In agreement with our observations, Hall et al. reported that BRAF-mutated melanoma cell lines have dysfunctional OXPHOS [53]. To reiterate, evaluation of the activity of OXPHOS complexes showed higher levels in BRAF-mutated and BRAF/NRAS wild-type melanoma cell lines compared to HDFs, most likely due to a higher mitochondrial mass as indicated by VDAC1 immunostaining and activity of citrate synthase. Several studies found that a BRAF mutation in melanoma causes increased glycolysis but also attenuates OXPHOS [12,21,57]. In contrast, our BRAF-mutated melanoma cell lines displayed high levels of glycolysis and OXPHOS complex activity. Melanogenesis, a multistep and highly regulated pathway, can switch metabolism of cells from OXPHOS to anaerobic glycolysis [27,28,58] and increase mitochondrial mass [59]. For the A375 and WM3311 melanoma cell lines comparable melanin levels have been reported [60,61]. Thus, the melanin content of these two cell lines cannot explain differences seen in OXPHOS activity, respiration, and glycolytic activity (Figure 1 and Figure 2).

Therapy resistance is a major clinical challenge and the resistance of melanoma to targeted therapy is accompanied by some distinct changes in cellular metabolism, most notably upregulation of OXPHOS [13,16,26,50,51]. In this context, our analysis of enzymatic activities of OXPHOS complexes shows upregulation of complex IV in a BRAF inhibitor-resistant melanoma cell line. 

Studies show that inducing OXPHOS impairment in cancer cells helps to suppress tumorigenesis [31,62]. We hypothesize that melanoma cells with dysfunctional OXPHOS should be more sensitive to drugs that target mitochondria. To address this hypothesis, we evaluated the effects of TIG, DOX, AZI, PP and ONC212 on melanoma cell lines versus HDFs, as non-cancerous cells.

The endosymbiotic hypothesis proposes that mitochondria originated from bacteria [63]. Therefore, some antibiotics such as those used in the present study (TIG, DOX and AZI) reduce mitochondrial and bacterial translation by binding to their ribosomes [31,38,64]. Inhibition of mitochondrial protein synthesis leads to reduced mitochondrial respiration and ATP production in mammalian cells, ultimately resulting in growth inhibition. Rapidly dividing cancer cells with high energy demands are more sensitive to antibiotics compared with non-cancerous cells, as confirmed in many in vitro and in vivo studies [31,38]. Similarly, in our study, HDFs, as non-cancerous skin cells, were less sensitive to antibiotics compared to melanoma cells. The anti-tumor effects of TGI, DOX and AZI on melanoma growth was reported previously [65,66,67]. However, here, for the first time, we compared these three antibiotics in parallel and investigated their effects on mitochondrial respiration in melanoma. We observed that TIG, DOX, and AZI have a variable effect on mitochondrial respiration and cell growth. In a side-by-side comparison, equal micromolar concentrations of AZI showed almost no effect on either cellular proliferation or mitochondrial respiration. The effect of DOX on proliferation was comparable to that of TIG, though surprisingly the BRAF-mutated melanoma cell line A375 showed resistance. In contrast to DOX, TIG significantly reduced the proliferation and respiration not only of melanoma cells but also of HDFs. The enhanced activity of TIG might be attributed to putative stronger binding affinity to the mitochondrial ribosomal subunit compared to DOX, as was shown in bacteria [64]. After treatment with TIG or DOX, the MAXR was consistently diminished in all melanoma cell lines, but interestingly not in HDFs, indicating lower tolerability to stress in melanoma in response to antibiotics, likely due to their dysfunctional mitochondrial respiration. Although the results of many in vitro and in vivo studies support the potent anticancer potential of antibiotics [31], a recent clinical study warned that antibiotics reduced the progression-free survival of patients treated with immune checkpoint inhibitors [68]. In addition, other clinical studies have reported that long-term use of antibiotics could increase the risk of cancer, likely due to attenuated immune surveillance, increased inflammation, and altered organ microbiota [69,70,71].

PP is used clinically for intestinal pinworm infections [72] and has recently gained attention as an anti-cancer drug [35,52,73,74,75]. In vitro and in vivo studies showed that PP inhibits proliferation of different types of cancer, including solid tumors and blood cancers, by reducing mitochondrial respiration [31,52,76,77]. In our study, we observed that PP is cytotoxic not only to melanoma cells but also to HDFs, by inducing OXPHOS dysfunction. PP-treatment unexpectedly caused an increased rate of OCR after injection of oligomycin, most likely due to an effect on mitochondrial membrane potential. Similarly, an increase of OCR after PP-treatment of myeloid leukemia in response to oligomycin was shown previously [35]. PP inhibits the NADH-fumarate reductase system, which alters the mitochondrial membrane potential. The NADH-fumarate reductase system is part of the OXPHOS system [77]. Mitochondrial membrane potential regulates respiratory rate and ATP synthesis and is itself generated by proton pumps (Complexes I, III, and IV) [78]. To date, severe side effects from oral administration of PP for treatment of gastrointestinal parasites have not been reported. This might be because of limited absorption of the drug into the bloodstream and subsequent low systemic concentration [79]. To target tumor cells, an anti-cancer drug needs to circulate in the blood to be able to reach the cancer cells. Therefore, we speculate that the oral use of PP might not effectively inhibit tumor growth, as it has been shown that oral administration of PP alone did not significantly reduce tumor growth in mouse models [73,80]. Since PP mainly remains in the intestine, its effect on intestinal cancers could be different, as also discussed by Li et al. [81]. In contrast, Esumi et al. argued that PP could be absorbed by the mammalian intestine and showed that orally administered PP reduced PANC-1 tumors in mice [82]. However, systemic circulation of PP might lead to substantial toxicity, based on its dramatic effects on growth and mitochondrial respiration of non-cancerous cells, as we observed. In agreement, two injections of 1 mg/kg PP caused the death of 5 of 9 mice [83]. However, a gradually increasing intraperitoneal PP dose reduced breast cancer growth without causing death in mice [83]. Drug-induced mitochondrial toxicity can further lead to liver, muscle, kidney and central nervous system failure [84]. 

According to the National Cancer Institute (USA) and several publications, the first-in-class analogue of the imipridone family, ONC201, is currently being tested for treatment of several types of solid tumors and hematological malignancies in clinical trials [39,83,85,86,87]. The anti-cancer activity of imipridones is linked to different mechanisms such as inactivation of Akt and ERK, activation of mitochondrial caseinolytic protease P (ClpP), protein quality control of the endoplasmic reticulum, the cellular stress response, apoptosis and cell-cycle arrest [40,44,88]. ONC212, another analogue of this family, displays enhanced anti-tumor effects in comparison with ONC201 [41]. Recently, ONC201 and ONC212 have been shown to reduce mitochondrial respiration in breast cancer, glioblastoma and lymphoma [42,43,44]. In agreement with these studies, we observed that ONC212 lowered mitochondrial respiration, including basal respiration, ATP-R and MAXR independent of the BRAF or NRAS mutation status. Interestingly, all melanoma cell lines were sensitive to ONC212 in terms of proliferation. This finding is in agreement with Wagner et al., who previously showed that 51 melanoma cell lines were sensitive to ONC212 [41]. By further investigating the underlying mechanisms of ONC212, we found that ONC212 directly reduces the enzymatic activity of OXPHOS complexes, apart from complex V, in melanoma cells. Interestingly, Ishizawa et al. similarly observed a reduction of OXPHOS complex activities, except complex V, through ClpP overexpression induced by ONC201 or ONC212 [44].

## 5. Conclusions

Based on our findings, melanoma cells can be categorized into two groups: one showing features of the classic Warburg effect and the other retaining a high level of respiration and OXPHOS complex activities despite being glycolytic. As we postulated, we found that melanoma cells have a lower capacity to resist the effects of mitochondrial targeting drugs, owing to their dysfunctional OXPHOS. Melanoma cell lines treated with TIG, DOX, or ONC212 showed reductions in MAXR, indicative of reduced respiratory capacity. Our study demonstrates that the effect of ONC212 is superior to that of antibiotics and PP, as it was the most effective and least toxic compound, and it also inhibited the growth of vemurafenib-resistant melanoma cells. Therefore, ONC212 may be a promising drug to be included into a multi-modal treatment regimen combining mitochondrial-targeted therapy with standard melanoma therapy. 

## Figures and Tables

**Figure 1 biomolecules-10-01395-f001:**
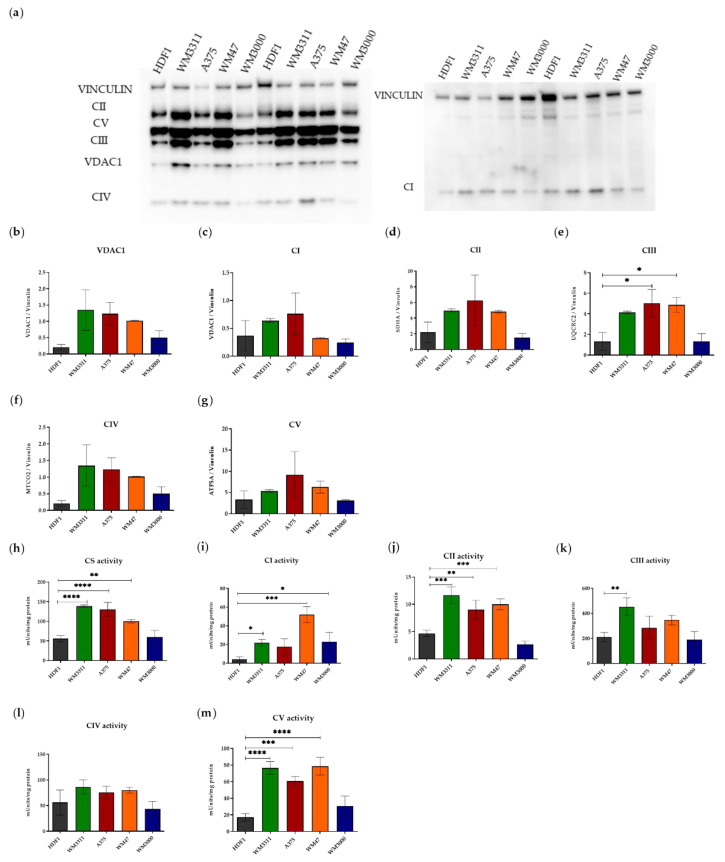
Melanoma cell lines show higher levels of OXPHOS compared to HDFs. (**a**) The left and right panels represent two separate western blots because of the equal size of the MTCO2 (CIV) and the NDUFS4 (CI) protein. (**b**–**g**) Western blot analysis for subunits of OXPHOS complexes (CI-CV), VDAC1 and vinculin in HDF1, WM3311, A375, WM47 and WM3000 cells. (**h**–**m**) Absolute activities of citrate synthase and OXPHOS complexes in HDF1, WM3311, A375, WM47 and WM3000 cells. Values are given as mean ± SD. One-way ANOVA followed by Dunnett’s Multiple Comparison Test; * *p* ≤ 0.05; ** *p* ≤ 0.01; *** *p* ≤ 0.001; **** *p* ≤ 0.0001. For western blot and enzymatic activity: post-nuclear supernatant containing the mitochondrial fraction was used. Two and three independent experiments were performed for western blot (n = 2) and enzymatic activity (n = 3), respectively. The levels of OXPHOS complexes and VDAC1 were normalized to vinculin. The activities of the OXPHOS complexes were normalized to protein concentration in cell lysates. Citrate synthase (CS); Complex I (CI); Complex II (CII); Complex III (CIII); Complex IV (CIV); Complex V (CV).

**Figure 2 biomolecules-10-01395-f002:**
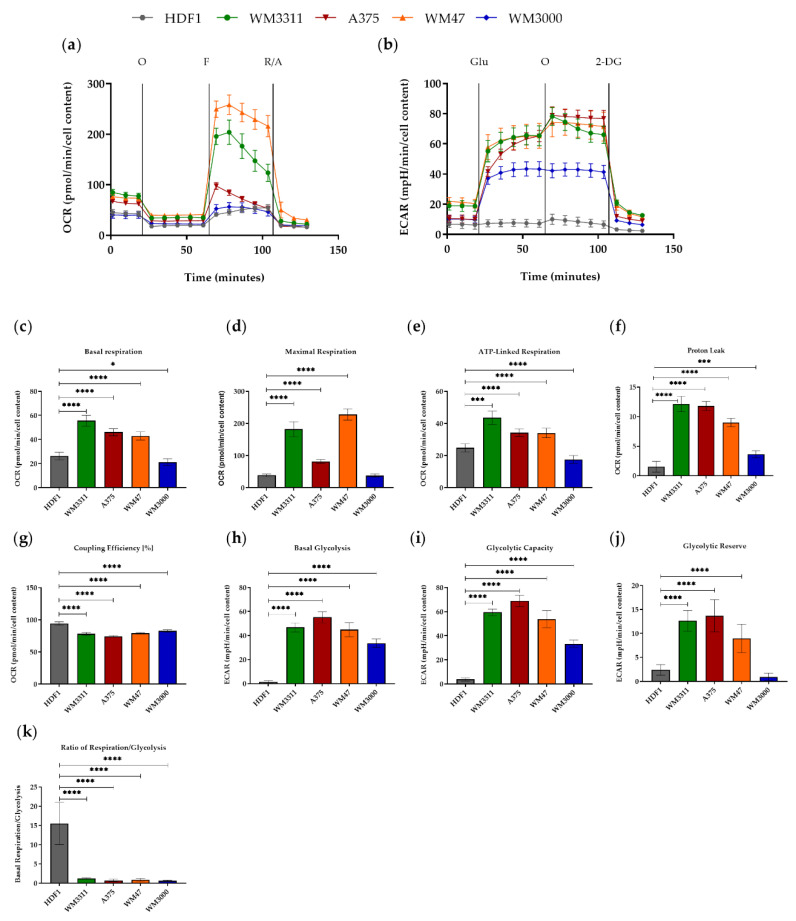
Results of the Mito Stress (**a**) and Glycolysis Stress (**b**) tests in HDF1, WM3311, A375, WM47 and WM3000 cells are presented as real-time measurements of OCR. (**c**) Basal respiration, (**d**) maximal respiration, (**e**) ATP-linked respiration, (**f**) proton leak, (**g**) coupling efficiency %, (**h**) basal glycolysis, (**i**) glycolytic capacity, (**j**) glycolytic reverse, (**k**) ratio of respiration to glycolysis. Values are given as mean ± SD. One-way ANOVA followed by Dunnett’s Multiple Comparison Test; * *p* ≤ 0.05; *** *p* ≤ 0.001; **** *p* ≤ 0.0001. For every cell line, two independent measurements with 6 to 8 replicates were performed. 10 mM Glucose (Glu); 5 µM Oligomycin (O); 2 µM FCCP (F); 0.5 µM Rotenone and Antimycin A (R/A); 50 mM 2-Deoxy-d-glucose (2DG).

**Figure 3 biomolecules-10-01395-f003:**
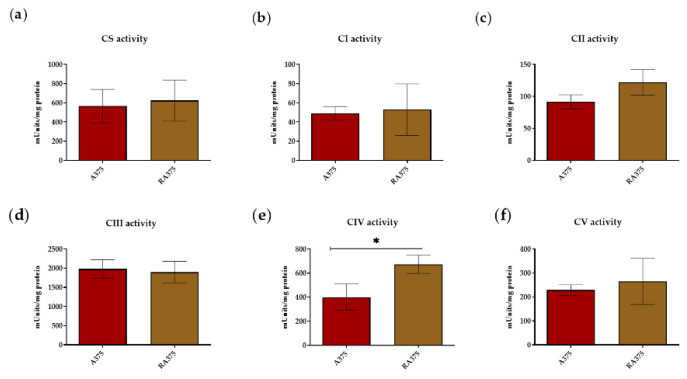
(**a**–**f**) Absolute activities of citrate synthase and OXPHOS complexes in isolated mitochondria in R-A375 and A375 cells. Citrate synthase (CS); Complex I (CI); Complex II (CII); Complex III (CIII); Complex IV (CIV); Complex V (CV). Values are given as mean ± SD. Unpaired *T*-test; * *p* ≤ 0.05. The activities of the OXPHOS complexes were normalized to the protein concentration in cell lysates. Measurements were repeated independently three times and each measurement included two replicates (n = 3).

**Figure 4 biomolecules-10-01395-f004:**
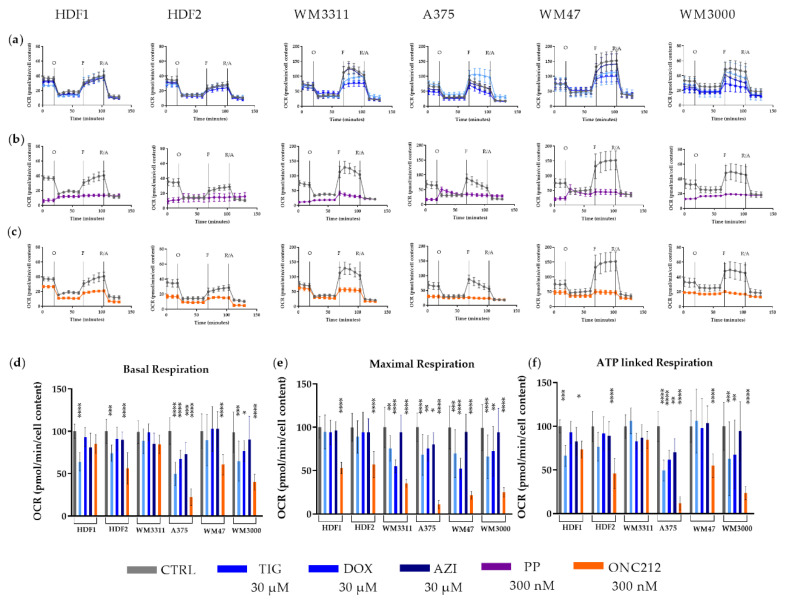
OCR measurements of (**a**) TIX-, DOX- and AZI-treated, (**b**) PP-treated, (**c**) ONC212-treated melanoma cells and HDFs compared to corresponding vehicle-treated cells during a mitochondrial stress test. Levels of (**d**) basal respiration, (**e**) maximal respiration, and (**f**) ATP-linked respiration in TIX-, DOX-, AZI- and ONC212-treated cells expressed relative to values of vehicle-treated cells normalized to 100%. Values are given as mean ± SD. One-way ANOVA followed by Dunnett’s Multiple Comparison Test; * *p* ≤ 0.05; ** *p* ≤ 0.01; *** *p* ≤ 0.001; **** *p* ≤ 0.0001. For every cell line, two independent measurements with 6 to 8 replicates were performed. Cells were treated with 30 µM TIG, DOX and AZI, and 300 nM PP and ONC212. 5 µM Oligomycin (O); 2 µM FCCP (F); 0.5 µM Rotenone and Antimycin A (R/A).

**Figure 5 biomolecules-10-01395-f005:**
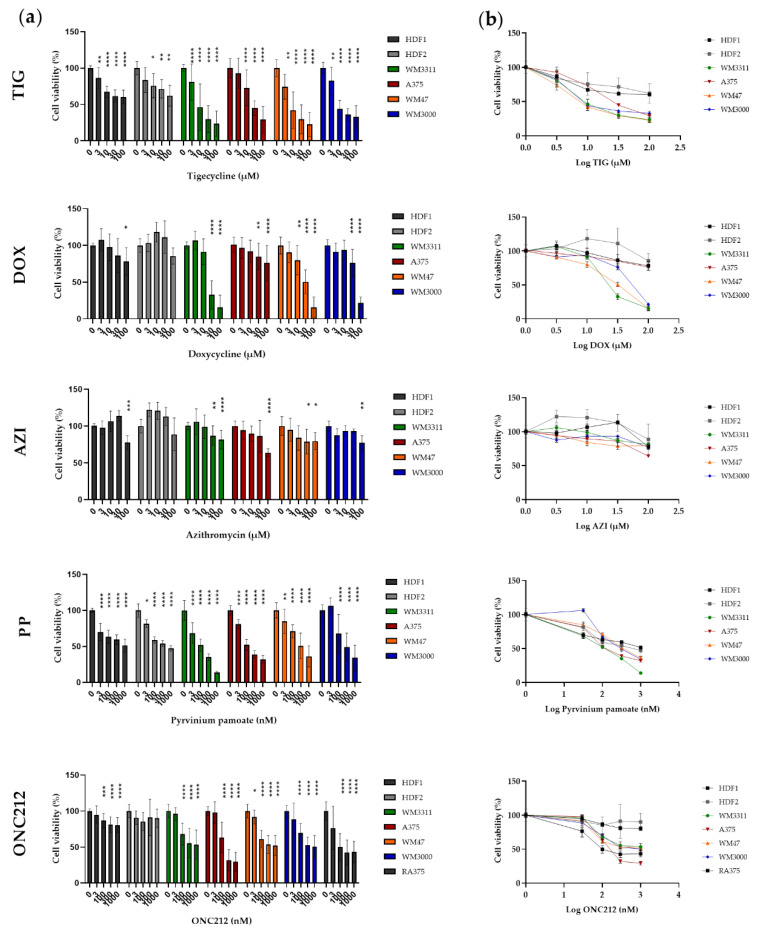
(**a**) Effect of TIG, DOX, AZI, PP and ONC212 on the growth of monolayer cultures of melanoma and HDF cell lines at the indicated concentrations for 72 h. (**b**) Dose response curve to generate IC50 values of drugs in melanoma cell lines and HDFs. Values are given as mean ± SD. Values are expressed relative to cell viability in vehicle-treated cells normalized to 100%. Values are given as mean ± SD. One-way ANOVA followed by Dunnett’s Multiple Comparison Test; * *p* ≤ 0.05; ** *p* ≤ 0.01; *** *p* ≤ 0.001; **** *p* ≤ 0.0001. Two to five independent experiments with four replicates were performed.

**Figure 6 biomolecules-10-01395-f006:**
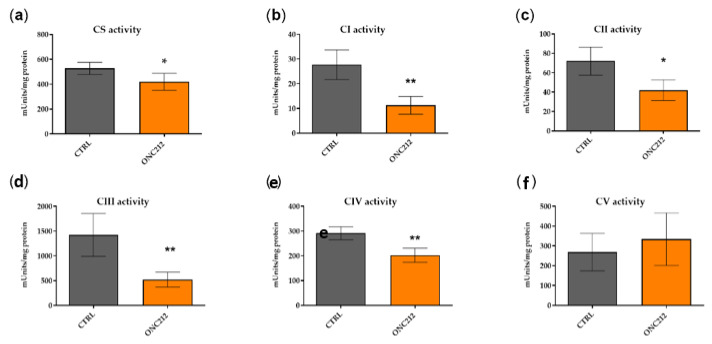
Absolute activities of citrate synthase and OXPHOS complexes in isolated mitochondria from 300 nM ONC212- or vehicle-treated A375 cells. (**a**) Citrate synthase (CS); (**b**) Complex I (CI); (**c**) Complex II (CII); (**d**) Complex III (CIII); (**e**) Complex IV (CIV); (**f**) Complex V (CV). Values are given as mean ± SD. Unpaired *T*-test; * *p* ≤ 0.05; ** *p* ≤ 0.01. The activities of OXPHOS complexes were normalized to protein concentration in cell lysates. Measurements were repeated three times and each measurement includes two replicates (n = 3).

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
