# Peer review of "Targeting Mitochondria in Melanoma"

_biomolecules, 2020, doi:10.3390/biom10101395_

Round 1

Reviewer 1 Report

This study demonstrates that ONC212 has an inhibitory effect on OXPHOS complex activity. Figure 1 may be revised to explain the differences between western blot bands of Vinculin in the left lane and the central lane in Fig1a. The reason why CII activity is higher in WM3311, A375 and WMA47 may be discussed. The number of samples may be added. The measurement of the enzymatic activities may be described more in detail in 2.4.

Author Response

Reviewer #1: This study demonstrates that ONC212 has an inhibitory effect on OXPHOS complex activity.

Figure 1 may be revised to explain the differences between western blot bands of Vinculin in the left lane and the central lane in Fig1a.

We thank the reviewer for raising this point. We clarified in the figure legend that two separate western blots were performed because complex IV and I have a similar molecular weight and therefore, cannot be analyzed on one blot. We revised the figure legend as indicated below (changes are indicated in red):

Figure 1. Melanoma cell lines show higher levels of OXPHOS compared to HDFs. (a) The left and right panels represent two separate western blots because of the equal size of the MTCO2 (CIV) and the NDUFS4 (CI) protein. (b-g) Western blot analysis for subunits of OXPHOS complexes (CI-CV), VDAC1 and vinculin in HDF1, WM3311, A375, WM47 and WM3000 cells. (h-m) Absolute activities of citrate synthase and OXPHOS complexes in HDF1, WM3311, A375, WM47 and WM3000 cells. Values are given as mean ± SD. One-way ANOVA followed by Dunnett’s Multiple Comparison Test; * p ≤ 0.05; ** p ≤ 0.01; *** p ≤ 0.001; **** p ≤ 0.0001. For western blot and enzymatic activity: post-nuclear supernatant containing the mitochondrial fraction was used. Two and three independent experiments were performed for western blot (n = 2) and enzymatic activity (n = 3), respectively. The levels of OXPHOS complexes and VDAC1 were normalized to vinculin. The activities of the OXPHOS complexes were normalized to protein concentration in cell lysates. Citrate synthase (CS); Complex I (CI); Complex II (CII); Complex III (CIII); Complex IV (CIV); Complex V (CV).

The reason why CII activity is higher in WM3311, A375 and WM47 may be discussed.

We thank the reviewer for the suggestion. We found higher activity not only of complex II, but of all OXPHOS complexes in WM3311, A375 and WM47 cells compared to HDFs. The reason is most likely due to higher mitochondrial mass as indicated by VDAC1 staining and activity of citrate synthase. Accordingly, we modified one sentence in the discussion part (changes are indicated in red):

To reiterate, evaluation of the activity of OXPHOS complexes showed higher levels in BRAF-mutated and BRAF/NRAS wild-type melanoma cell lines compared to HDFs, most likely due to a higher mitochondrial mass as indicated by VDAC1 immunostaining and activity of citrate synthase.

The number of samples may be added.

Thank you for your kind reminder, we added the number of samples in the legends of figures 1, 3 and 6.

The measurement of the enzymatic activities may be described more in detail in 2.4

We now describe the methods for measuring OXPHOS enzymatic activities more in detail in section 2.4. (changes are indicated in red)

Citrate synthase (CS) and OXPHOS enzyme activities were determined by spectrophotometric measurements as previously described [39,40]. The 600 g homogenate used to measure OXPHOS enzyme activities in four melanoma cell lines and one HDF line. Furthermore, isolated mitochondria from RA375, A375, ONC212-treated A375 and vehicle treated-A375 pellets were used for enzymatic measurements. Mitochondria were isolated according to Bentlage et al. [41]. In brief, citrate synthase was measured according to Srere et al. [43] with modifications. The reaction mixture contained 50 mM Tris-HCl pH 8.1, 0.1% bovine serum albumin (BSA), 0.1% Triton X-100, 0.2 mM 5,5’-dithio-bis (2-nitrobenzoic acid) and 0.15 mM acetyl-CoA, and 10 µl of 600 g homogenate or 2 µl of isolated mitochondria. After initially recording thiolase activity for 3 minutes (min), the citrate synthase reaction was started by addition of 0.5 mM oxaloacetate and was followed at 412 nm for 8 min.

The activity of NADH/decylubiquinone oxidoreductase (complex I) was measured by monitoring NADH oxidation in the presence of quinone at 340 nm for 4 min. After 3 min pre-incubation of samples (10 µl of 600 g homogenate or 10 µl isolated mitochondria) at 37 degrees Celsius, the reaction was started by the addition of 0.2 mM NADH. Complex I activity was calculated using the difference between the measured rates in the absence vs. presence of 0.5 mg/ml rotenone, an inhibitor of complex I.

The activity of succinate:ubiquinone-oxidoreductase (complex II) was measured according to Rustin et al. [44] with the following modifications. The reaction mixture contained 50 mM potassium phosphate pH 7.8, 2 mM ethylenediaminetetraacetic acid (EDTA), 0.1% BSA, 3 µM rotenone, 80 µM 2,6-dichlorophenol, 50 µM decylubiquinone, 1 µM antimycin A, 0.2 mM ATP, 0.3 mM KCN, and 10 µl of 600 g homogenate or 2 µl of isolated mitochondria. The mixture was preincubated for 10 min at 37 degrees Celsius, the reaction was started by addition of 10 mM succinate, and followed for 6 min at 600 nm.

The activity of coenzyme Q/cytochrome c—oxidoreductase (complex III) was determined in the presence of 50 mM potassium phosphate buffer pH 7.8, 2 mM EDTA pH 8.6, 0.3 mM KCN, 100 µM cytochrome c, 200 µM reduced decyl-ubiquinol. The reaction was started by the addition of samples (10 µl of 600 g homogenate or 2 µl of isolated mitochondria). The reduction of cytochrome c was monitored over 3-4 min at 550 nm. The reaction was inhibited by the addition of 1 μM antimycin. Antimycin A-insensitive activity was subtracted from the total activity to calculate complex III activity.

The enzyme activities of ferrocytochrome c/oxygen oxidoreductase (complex IV), and the oligomycin-sensitive ATPase activity of the F1F0 ATP synthase (complex V) were determined by using buffer conditions as previously described by Rustin et al. [44]. The complex IV activity was measured by following the oxidation of cytochrome c at 550 nm for 6 min. 10 µl of 600 g homogenate or 2 µl of isolated mitochondria were used. The whole reaction mixture for the ATPase activity measurement was treated for 10 seconds with an ultra-sonifier (Bio cell disruptor 250, Branson, Vienna, Austria). The complex V activity was measured by following the oxidation of NADH at 340 nm for 4 min. 7 µl of 600 g homogenate or 2 µl of isolated mitochondria were used. All spectrophotometric measurements (Uvicon 922, Kontron, Milan, Italy) were performed at 37 degrees Celsius. All reagents were obtained from Sigma Aldrich.

Reviewer 2 Report

This is an interesting paper. Methodology and data collection and interpretations appear to be appropriate. However, final version would benefit from some improvements.

Melanin synthesis is an important phenotypic feature affecting behavior and metabolism of normal and malignant melanocytes (Physiol Rev 84, 1155-1228, 2004; Pigment Cell Melanoma Res 25, 14-27, 2012; Exp  Dermatol 24: 258-259, 2015).

What was the phenotype of the tested melanoma lines?

Note that melanogenic apparatus can also affect outcome of melanoma therapy.

In melanomas changes in HIF-1a protein expression and subcellular localization are coupled to changes in energy yielding metabolism (Arch Biochem Biophys 563:79-93, 2014).

Why this important transcriptional regulator was not evaluated?

Why dermal fibroblasts but not epidermal melanocytes are used as examples of normal skin cells?

The WM lines are from Wistar Institute. Were they provided by Dr Meenhard Herlyn? If yes, this should be acknowledged

Author Response

Reviewer #2: This is an interesting paper. Methodology and data collection and interpretations appear to be appropriate. However, final version would benefit from some improvements.

Melanin synthesis is an important phenotypic feature affecting behavior and metabolism of normal and malignant melanocytes (Physiol Rev 84, 1155-1228, 2004; Pigment Cell Melanoma Res 25, 14-27, 2012; Exp Dermatol 24: 258-259, 2015). What was the phenotype of the tested melanoma lines? Note that melanogenic apparatus can also affect outcome of melanoma therapy.

We thank the reviewer for raising this interesting point. Unfortunately, we did not measure the level of melanin in our melanoma cell lines and we did not find a reference, which compares the level of melanin among the melanoma cell lines WM3311, WM47, WM3000 and A375. However, we will consider the level of melanin in melanoma cell lines in our future studies.

In melanomas changes in HIF-1a protein expression and subcellular localization are coupled to changes in energy yielding metabolism (Arch Biochem Biophys 563:79-93, 2014). Why this important transcriptional regulator was not evaluated?

We agree that transcription of HIF-1a plays important roles in cancer metabolism. Since we previously did not observe differences in the level of HIF-1a among melanoma tissues (Ref. 8), we did not expect a remarkable difference among our melanoma cells. In addition, the expression of HIF-1a is mostly altered by hypoxic conditions; therefore, we did not focus on HIF-1a in the present study.

Why dermal fibroblasts but not epidermal melanocytes are used as examples of normal skin cells?

We fully agree with the reviewer that epidermal melanocytes are a better control for melanoma cells. We tried to amplify melanocyte cells purchased from Sigma or primary melanocyte cells in different recommended cell culture media. However, we were not able to reach sufficient numbers of melanocytes for enzymatic activity or bioenergetics measurements.

The WM lines are from Wistar Institute. Were they provided by Dr Meenhard Herlyn? If yes, this should be acknowledged.

We thank the reviewer for the kind reminder. We now ackowledge Dr Meenhard Herlyn in section 2.1, with the following sentence: The melanoma cell lines WM3311 and WM47 were kindly provided by Dr. Meenhard Herlyn, Wistar Institute.

Round 2

Reviewer 1 Report

The authors have revised the manuscript.

Author Response

There were no further points to address

Reviewer 2 Report

The authors inadequately replied to the critique and have not made any relevant changes in the manuscript.

Melanin pigmentation with relevant citations and HIF-1 expression seen by others should be mentioned both in the introduction and Discussion

Note using melanocytes should be commented in the section limitation of the study

Round 3

Reviewer 2 Report

The authors adequately revised the ms